# Impact Abrasive Wear Property of CrAlN/TiSiN Multilayer Coating at Elevated Temperatures

**DOI:** 10.3390/ma15062214

**Published:** 2022-03-17

**Authors:** Ying Luo, Yuanyuan Dong, Cong Xiao, Xiaotong Wang, Hang Peng

**Affiliations:** 1School of Chemical Engineering and Technology, Tianjin University, Tianjin 300072, China; 2Science and Technology on Reactor System Design Technology Laboratory, Nuclear Power Institute of China, Chengdu 610041, China; sshuimus@163.com (Y.D.); king19870101@163.com (C.X.); xtwang129@gmail.com (X.W.)

**Keywords:** abrasive particle, impact wear, CrAlN/TiSiN multilayer coating, elevated temperature

## Abstract

The impact wear property of hard coatings at elevated temperatures is of particular interest for applications in nuclear power plants. This study evaluated the impact wear behavior of two CrAlN/TiSiN coatings with and without sand. Alternately grown CrAlN and TiSiN films with modulation periods of 455 and 19 nm were formed in a columnar structure. The nanomultilayer shows better impact wear resistance than multilayer films with and without sand. The energy absorption rate has a similar trend to wear rate, leading to lower rebound velocity and peak impact force of the nanomultilayer compared with that of the multilayer. CrAlN/TiSiN coatings can protect the 308L substrate from oxidation. The dominant impact wear mechanism without sand is plastic deformation, and this wear region can be defined as the percussive zone. Peeling occurs on the multilayer surface without sand after 10^4^ percussions, leading to rapid oxidation of the 308L substrate at 500 °C. Due to the abrasion effect, the wear rate of the sample with sand increases by an order of magnitude compared to the sample without sand. The wear scar of the sample with sand can be divided into the mixing zone and the sand−affected zone from inside to outside. Fe oxides are formed beyond the unbroken coating, which may be related to the outward diffusion of Fe.

## 1. Introduction

In nuclear power plants, the heat transfer system is susceptible to various types of impact wear modes, such as percussive impact or particle erosion [1,2]. Percussive impact wear usually occurs at the grid/rod interface of fuel cladding and the tube/anti−vibration bars of steam generators [3]. Solid particles may be generated from processing defects, corrosion products, and foreign particles [4], causing material removal during the normal operation of a nuclear power plant [5,6]. Severe impact wear and material removal are responsible for the thinning of a heat transfer tube, leading to coolant leakage and catastrophic safety problems. Hard coatings are ideal candidates for applications in nuclear power plants because of their chemical inertness at evaluated temperatures [7,8].

Physical vapor deposition (PVD) is an effective preparation method for hard coatings [9,10]. In particular, CrAlN and TiSiN ternary nitride coatings are widely used, due to their excellent functional properties of high temperature oxidation and wear resistance [11]. The oxidation resistance of the CrAlN coating is mainly ascribed to the formation of a compact Al_2_O_3_ film, which can prevent the outward diffusion of Al and the inward diffusion of O [12]. The wear resistance of the TiSiN coating is outstanding because of the formation of amorphous Si_3_N_4_ [13,14]. The crystal TiN phase is surrounded by Si_3_N_4_, resulting in refinement of grains and enhancement of hardness. However, a monolayer coating has high residual stress and poor toughness, which limit its application as a protective coating [15].

Hard coatings with multilayer structures have been studied to further improve their performance [16,17]. A multilayer coating shows remarkable advantage to improve comprehensive properties, such as adhesion and toughness [18,19]. Zhang et al. [20] reported that an AlTiN/AlCrSiN multilayer coating with 8.3 nm modulation period presented the highest wear resistance at the evaluated temperatures. The oxidation resistance substantially increased via the incorporation of AlCrSiN. Miletić et al. [21] found that the wear rate of a nanolayer CrAlN/TiSiN coating reduced by an order of magnitude, and the stable coefficient of friction was four times lower than that of the monolayer TiSiN coating. Alat et al. [22] studied the corrosion property of a multilayer coating in pure water (350 °C, 18.7 MPa) for 90 days. No spalling and delamination occurred on the multilayer coating. Multilayer structures improve the performance of hard coatings in extreme and demanding applications [23,24]. However, the anti−wear of hard coatings is usually evaluated by sliding wear tests, and the impact abrasive wear performance is always neglected. Impact abrasive wear, as typical wear modes in nuclear power plants, should be studied on its candidate materials.

In this paper, two different types of CrAlN/TiSiN coatings were designed and their impact wear behavior at elevated temperatures was investigated, with a focus on the differences in multilayer and nanomultilayer. Impact wear tests were performed with and without sand, and the coating structure, interface response, and wear scar morphology were characterized. It is helpful to establish a more comprehensive evaluation method and provide candidate materials for nuclear power plants.

## 2. Materials and Methods

### 2.1. Coating Deposition Process

CrAlN/TiSiN multilayer and nanomultilayer coatings were deposited on 308L stainless steel substrate with a dimension of 30 mm × 30 mm × 2 mm through multi-arc ion plating. This arc−PVD system was designed for industrial production and equipped with direct current power sputtering of TiSi and CrAl targets. Two pairs of TiSi and CrAl targets were used in the coating system and the same target was placed in opposition. Before deposition, the surface of 308L substrate was ground by 600#, 1000#, 1500#, and 2000# sandpaper and ultrasonically cleaned in absolute ethanol and petroleum ether for 15 min. Then, the substrates were fixed on the sample holder. The rotation speed of the sample holder was 3 r/min. The pressure and temperature in the chamber were set to 10^−3^ Pa and 450 °C, respectively. High−purity Ar was used to etch the substrate to eliminate impurities on the surface of the substrate at a bias voltage of 200 V. N_2_ was injected into the chamber, and the coating was deposited at a working pressure of 4 Pa. The bias voltages for multilayer and nanomultilayer were 40 and 100 V, respectively. The target currents on TiSi and CrAl were 120 and 150 A, respectively. The deposition times for the multilayer and nanomultilayer coatings were 180 and 240 min, respectively.

### 2.2. Impact Wear Test with Abrasive Particle

Quartz sand was selected as the abrasive particle during impact wear testing to simulate particle erosion. The surface morphology and the X-ray diffractometer (XRD) pattern of quartz sand are presented in Figure 1. The particle size was filtered between 470 and 830 μm. The structure of quartz sand is mainly composed of SiO_2_ (Figure 1b).

The schematic diagram of the home−made impact wear tester is shown in Figure 2. A voice coil motor drove the mass block and Si_3_N_4_ ball with a diameter of 4.76 mm by contact tie rod to collide with the coating specimens. The mass block was released by the tie rod when the set velocity was reached. The Si_3_N_4_ ball rebounded freely on a linear guide after contact with the coating specimen. Repeated percussive wear at fixed energy occurred on the coating specimens. Contact force and displacement were measured by a force sensor and a displacement sensor. The coating specimen was placed in a heater to achieve impact wear at various temperatures. A water coolant system was installed between the force sensor and the coating fixture to avoid conducting heat to the force sensor. The quartz sand fell under the action of gravity and eroded the coating surface by coupling with percussive wear.

Velocity (v_i_) was calculated by dividing relative displacement (∆d_i_) by time interval (∆t_i_). Impact kinetic energy (E_i_), rebound energy (E_r_), and absorbed energy (E_a_) were obtained by kinetic energy calculation formula at various cycles. Energy absorption rate (σ) was calculated by ratio of absorbed energy and impact kinetic energy. The detailed impact wear parameters with sand and without sand are listed in Table 1.

### 2.3. Microstructural Observation

Scanning electron microscope (SEM) (Tescan Mira 3 XH, Brno, Czech Republic) and depth of field optical microscope were used for morphological observation of the coating structure and the wear scar. Coating composition was analyzed by energy dispersive spectrometry (EDS) and XRD. The scan angle ranged from 20° to 90° at a scan velocity of 5°/min. The nanohardness and elastic modulus of deposited coatings were measured by a nanoindentor. Transmission electron microscopy (TEM) (Talos F200S, Waltham, MA, USA) after focused ion beam (FIB) thinning was conducted to investigate the microstructure and element distribution of the CrAlN/TiSiN nanomultilayer. Wear volume and wear depth were determined using depth of field optical microscope.

## 3. Results and Discussion

### 3.1. Structure and Composition of CrAlN/TiSiN Coating

The chemical compositions of the CrAlN/TiSiN multilayer and nanomultilayer coatings are listed in Table 2. The two coatings have similar elemental contents. Figure 3 shows the XRD pattern of the two CrAlN/TiSiN coatings. The CrAlN/TiSiN multilayer and nanomultilayer coatings have the diffraction peaks of fcc–CrN, fcc–TiN, and fcc–AlN phases, with preferred orientation on the (200) crystal plane, (311) crystal plane, and (102) crystal plane, respectively. The diffraction peak of Si_3_N_4_ was detected in the CrAlN/TiSiN nanomultilayer coating, which may be related to the thinner coating compared with that of the multilayer structure. As reported in Ref. [25], SiN_x_ is formed between adjacent TiN nano−crystallites when the TiSiN film presents the superhardness effect. The shift of the AlN phase could be ascribed to the solid solution of Al with smaller atomic size in the Ti–N lattice [20].

Figure 4 shows the surface and cross−sectional morphologies of the CrAlN/TiSiN coatings with various modulation periods. The thicknesses of multilayer and nanomultilayer are 4.97 and 5.26 μm, respectively. Only fine droplets and pits were found on the surfaces of the two coatings, indicating that the microstructure deposited by PVD is relatively dense. Distinct modulation periods were found on the multilayer according to element distribution of Ti, N, Al, Cr, and Fe (Figure 4a). The 308L stainless steel substrate is enriched with Fe, while the two coatings are enriched with N. The transition layer is also enriched with Al and Cr, indicating that AlN and CrN is the transition layer. The AlN and CrN transition layer for the multilayer and nanomultilayer samples has thicknesses of 1.40 and 0.66 μm, respectively. Moreover, Cr and Ti in the multilayer are alternating in the outer layer. However, no obvious alternating distribution of Cr and Ti was detected in the nanomultilayer, which may be related to the limited magnification of SEM.

The EDS line scan profile from the surface to the inside is shown in Figure 4b to clarify the modulation period and layer number in the multilayer. The contents of Ti and Al are higher in the CrAlN film, while those of Ti and Si are higher in the TiSiN film. The modulation period and layer number are 7 layers and 455 nm, respectively.

The cross−sectional morphology of the CrAlN/TiSiN nanomultilayer was observed by TEM to explore microstructure (Figure 5). A typical columnar structure was found in the CrAlN/TiSiN nanomultilayer (Figure 5a). The higher magnification image and EDS mapping (Figure 5b) further reveal that the columnar grain is composed of nanomultilayers, which are parallel to the coating surface. The contents of Ti, Si, Cr, and Al fluctuate in the nanomultilayer, while the level of N is relatively stable in all films.

The EDS line scan profile in Figure 5c displays the alternating distribution of Ti, Si, Cr, and Al. The contents of Ti and Si are higher in the TiSiN film, while those of Cr and Al are higher in the CrAlN film. The modulation period of the CrAlN/TiSiN nanomultilayer is about 19 nm. The high−resolution transmission electron microscopy (HRTEM) images also show that the nanomultilayer structure consists of TiSiN film (dark zone) and CrAlN film (bright zone). Part of the amorphous zone was detected in the TiSiN film, which may be related to the existence of SiN_x_ [15,26]. The indexed selected−area electron diffraction (SAED) image in Figure 5e presents polycrystalline diffraction rings with randomly distributed spots, indicating that this coating is composed of fcc single−phase solid solution [27]. This finding is similar to the XRD pattern.

Figure 6 shows the nanohardness and elastic modulus of the two CrAlN/TiSiN coatings. The multilayer coating has a shallower indentation than the nanomultilayer coating. The measured nanohardness values of the multilayer and nanomultilayer coatings are 35.70 and 41.59 GPa, respectively. The nanohardness remarkably increases as the modulation period reduces to 10 or few nanometers [13,28]. The stress required for dislocation to slide between the layers with various shear moduli or to move the original dislocation within the layer increases with decreasing modulation period [29]. The formation of coherent/semi−coherent interfaces creates alternating strain fields in the multilayer coating, which impedes the dislocation movement [30]. This situation may prove that the nanomultilayer has higher wear resistance. Plasticity index (H^3^/E^2^) is also used to evaluate the resistance of material to plastic deformation [31,32]. The plasticity index of CrAlN/TiSiN coating increases from 0.56 GPa to 0.63 GPa as the modulation period decreases from 455 nm to 19 nm, indicating the improved toughness and plastic deformation ability from the superlattice effect [20]. A higher plasticity index usually indicates harder plastic deformation ability and lower material removal capacity, which may be more suitable to evaluate impact wear resistance.

### 3.2. Analysis of Interface Response

The complete velocity response of the two CrAlN/TiSiN coatings at 10^4^ cycles is shown in Figure 7. The rebound velocity of the nanomultilayer coating without sand is higher than that of the multilayer coating. This phenomenon is more obvious under sand conditions. In general, absorbed energy mainly transfers to material removal and plastic deformation [33,34,35]. A higher rebound velocity in the nanomultilayer coating indicates higher impact wear resistance. As reported by Ref. [35], sand can promote material removal due to the abrasion effect during the impact wear process. Therefore, higher kinetic energy is absorbed, and the rebound velocity is lower under sand conditions.

Figure 8 shows the impact force responses over the entire 10^4^ cycles. The peak impact force of the two CrAlN/TiSiN coatings under sand conditions is low and fluctuating compared with that of the coatings under no−sand conditions (Figure 8a,b). Sand with acute angle may easily cut into the coating when the Si_3_N_4_ ball comes into contact with the sand during impact testing. The contact time is longer at this time (Figure 8c,d). According to the momentum theorem, the peak force is definitely lower due to the lower rebound velocity. The upper deviation of the partial peak impact force is higher than that under the no−sand condition, which may be ascribed to broken sand. Broken sand with high hardness generally has brittle fracture [36], leading to higher peak impact force. The nanomultilayer has lower peak impact force than the multilayer.

### 3.3. Analysis of Impact Wear Mechanism

Figure 9 shows the surface morphology of the wear scar with and without sand after 10^4^ cycles. No obvious crack exists in the wear region, proving the excellent tolerance of the two coatings to crack propagation [37]. The appearance of black matter on the multilayer may manifest broken coating and exposure of 308L substrate (Figure 9a). This finding was not found in the nanomultilayer, indicating its better binding ability. Free falling sand at a certain speed come into contact with the Si_3_N_4_ ball and then collide with the coating surface under the sand condition. Therefore, the wear area of the coating with sand is four times that of the coating without sand. The strong abrasion effect may completely remove the coating to expose the 308L substrate. The wear area of the nanomultilayer coating is smaller than that of the multilayer coating, indicating the higher impact wear resistance of the former with sand and without sand.

Figure 10 shows the wear scar profiles of the two CrAlN/TiSiN coatings under sand or no−sand conditions. The edge of the wear scar under no−sand conditions is over the original profile, but no wear debris covers (Figure 9a,b), which is ascribed to plastic deformation caused by repeated percussion effect under elevated temperatures. The 308L substrate becomes soft at 500 °C, and part of the materials are squeezed to the edge of the wear scar, causing changes in the wear scar profile [38]. However, no obvious coating peels off, indicating the preeminent binding ability of the two coatings. The total wear scar of the coating without sand can be described as the percussive zone (blue dotted line, Figure 10b,c), which mainly suffers from plastic deformation. The wear scar of the coating with sand can be divided into the mixing zone (yellow dotted line) and the sand−affected zone (red dotted line) based on the surface morphology (Figure 10d,e). In the sand−affected zone, the coating surface mainly suffers from the abrasion effect from particle erosion and numerous pits are formed on the surface. These pits have shallower depth and fluctuate in the original surface. The wear depth in the mixing zone is higher than that in the percussive zone or sand−affected zone due to the synergistic effect of percussion and particle erosion. The higher wear profile at the edge of mixing zone may be related to plastic deformation and wear debris accumulation. The wear depth of the nanomultilayer coating is lower than that of the multilayer coating, especially under the sand condition. The nanomultilayer coating presents better impact wear resistance against percussion and particle erosion than the multilayer coating.

The calculated wear rate and energy absorption rate are shown in Figure 11. The wear rate of the two coatings with sand reaches 10^−3^ mm^3^/J, which is one order of magnitude higher than that without sand. Particle erosion brings more serious harm than percussion. Similar to the results of the wear scar profiles in Figure 10, the CrAlN/TiSiN nanomultilayer coating presents better impact wear resistance under no−sand or sand conditions. Wear rate has a similar trend as energy absorption rate because absorbed energy is mainly transferred to material removal and plastic deformation.

Figure 12 shows the wear scar morphology and elemental distribution of the two CrAlN/TiSiN coatings without sand to analyze impact wear property. The dominant impact mechanism without sand is plastic deformation. The elastic recovery of the coating after the 308L substrate plastic deformation will generate tensile stress [37]. Repeated tensile stress may cause coating peeling. The cross−sectional morphology in Figure 4 and Figure 5 indicates that the outer layer and transition layer of the CrAlN/TiSiN coatings are enriched with N, which can act as evidence for the existence of the coating. For the multilayer coating, Ti and N are deficient and Fe is enriched in corresponding positions, proving that part of the multilayer is completely peeled off from the 308L substrate. The substrate is exposed and quickly oxidized due to lack of coating protection. The chemical composition analysis at points A and B in Table 3 proves this phenomenon. However, in the wear scar, the concentration of O was detected not only where coating was peeled off, which may be related to the outward diffusion of Fe [39]. This phenomenon is obvious in the wear scar of the CrAlN/TiSiN nanomultilayer coating. The contents of Ti and N are similar inside and outside the wear scar, indicating that no peeling off occurs on the wear scar surface of the nanomultilayer coating (Figure 12b). However, Fe is concentrated in the wear scar. The chemical composition analysis at points C and D in Table 3 indicates that the coating still exists below the oxides. Percussion causes thinning of the coating, thereby reducing the diffusion time. The formed pitting during coating preparation provides pathways for Fe diffusion. Finally, Fe was detected in the oxide beyond the surface of the coating.

The design of the multilayer coating reduces its brittleness, which can effectively improve the tolerance to percussion [37]. The smaller modulation period of the multilayer can further improve the resistance of the coating to plastic deformation. As a result, no coating peeling occurs on the nanomultilayer, and local peeling occurs on the multilayer. The nanomultilayer has better binding ability than the multilayer. In addition, no O concentration was detected outside the wear scar, demonstrating that complete CrAlN/TiSiN coating has excellent high temperature oxidation resistance.

Figure 13 shows the wear scar morphology and elemental distribution of the two CrAlN/TiSiN coatings with sand. Obvious cutting and ploughing track were found in the sand−affected zone, resulting in the complete removal of the outer layer and transition layer of the two coatings. Some sand particles are embedded in the wear scar (Figure 13b). Therefore, Fe and Si are enriched in the wear scar, while N is deficient. The dominant impact wear mechanism of the two coatings with sand is the abrasion effect. The Fe content in the mixing zone in the center of the wear scar is lower than that in the sand−affected zone. Sand with large size is broken under percussive action, and numerous fine sand particles are embedded in the substrate. Although broken sand absorbs part of kinetic energy, abrasion causes more material removal by the same energy. The wear rate with sand is higher than that without sand.

The cross−sectional morphology of the nanomultilayer is presented in Figure 14 to further illustrate the effect of sand under impact process. In the sand−affected zone, sand with a certain velocity will cut the coating and substrate, causing obvious ploughing track and rough surface (Figure 14b). In the mixing zone, sand is broken and embedded into the material surface due to the effect of percussion. A small amount of residual coating is still attached to the substrate, indicating the occurrence of severe plastic deformation on the material surface. The surface of the 308L substrate also accumulates a layer of sand, according to the EDS mapping in Figure 14c. This finding is ascribed to repeated percussion. The thick sand layer causes lower Fe content in the center of the wear scar than that in the sand−affected zone (Figure 13).

## 4. Conclusions

CrAlN/TiSiN coatings were deposited on 308L stainless steel by magnetron sputtering system, and CrN was chosen as the transition layer. The alternate growth of CrAlN and TiSiN in the two coatings resulted in multilayer and nanomultilayer with modulation periods of 455 and 19 nm, respectively. Impact wear property with and without sand was evaluated on the two coatings at 500 °C. The following conclusions were drawn:The two CrAlN/TiSiN coatings present favorable tolerance to crack propagation during repeated percussion. The dominant impact wear mechanism without sand is plastic deformation, and the whole wear scar is defined as the percussive zone.Sand has an obvious abrasion effect on the coating surface, resulting in an increase in the wear rate of the coating with sand by an order of magnitude than that under the no−sand condition. The whole wear scar with sand can be divided into the mixing zone and the sand−affected zone from inside to outside.The peak impact force and rebound velocity under the sand condition are lower than under the no−sand condition due to higher energy absorbed for material removal.The nanomultilayer coating shows better plasticity index and impact wear resistance than the multilayer coating. In particular, no obvious peeling occurs in the nanomultilayer coating after 10^4^ cycles of percussion under the no−sand condition. Thinning coating and pits may cause the outward diffusion of Fe at evaluated temperatures, leading to the formation of oxides beyond the unbroken coating.

## Figures and Tables

**Figure 1 materials-15-02214-f001:**
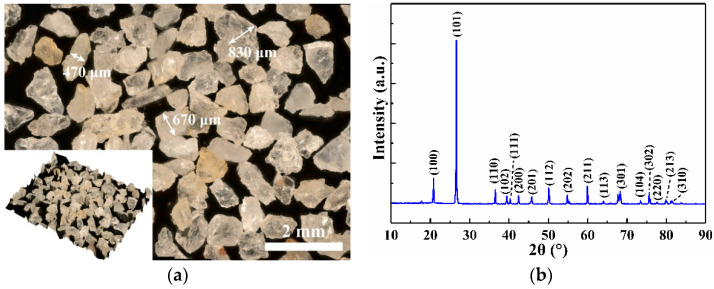
Characteristics of abrasive sand particle, (**a**) Surface morphology, (**b**) XRD pattern.

**Figure 2 materials-15-02214-f002:**
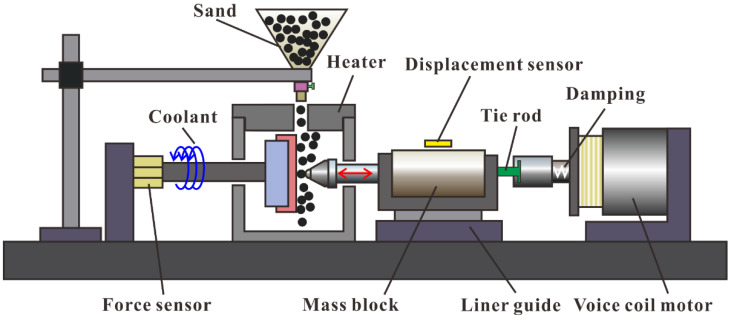
Schematic diagram of impact wear tester with sand at elevated temperatures.

**Figure 3 materials-15-02214-f003:**
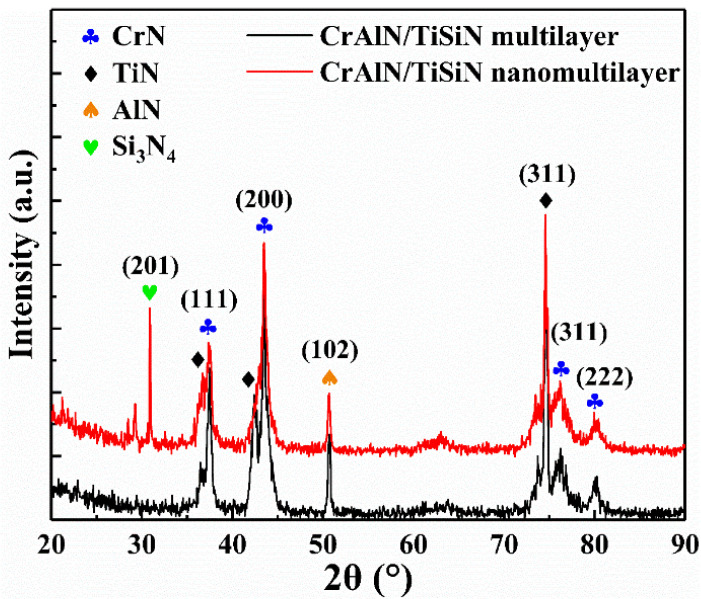
XRD patterns of CrAlN/TiSiN multilayer and nanomultilayer.

**Figure 4 materials-15-02214-f004:**
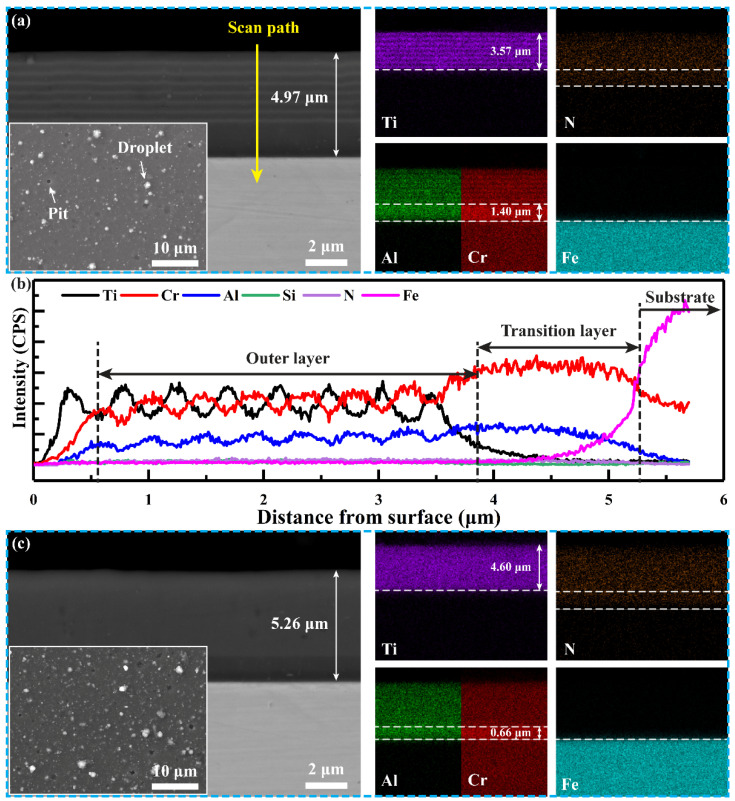
Surface and cross−sectional morphologies of CrAlN/TiSiN coatings with various modulation periods: (**a**) Multilayer; (**b**) EDS line scan along scan path in (**a**); (**c**) Nanomultilayer.

**Figure 5 materials-15-02214-f005:**
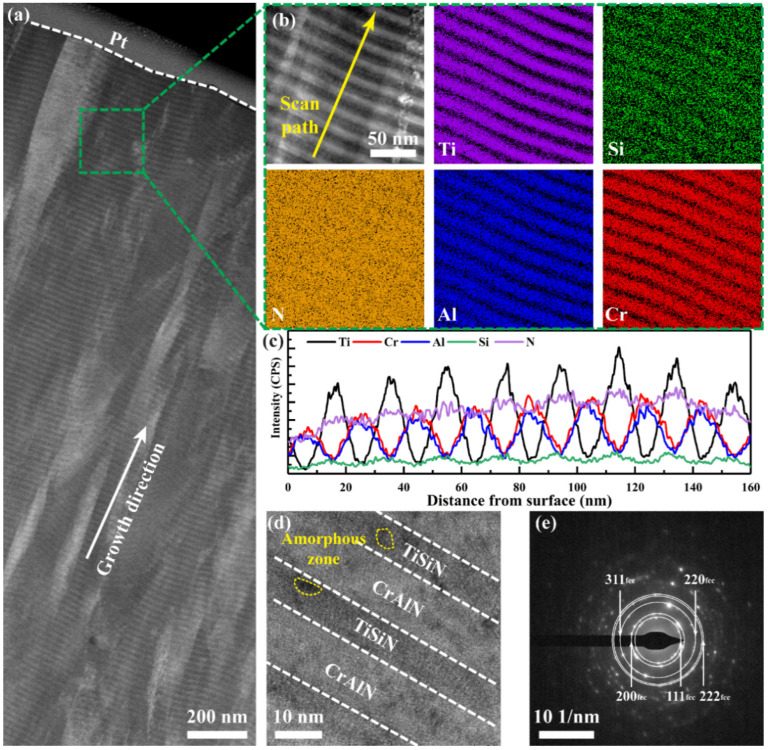
Cross–sectional TEM morphology of CrAlN/TiSiN nanomultilayer: (**a**) Low magnification TEM image; (**b**) EDS mapping of typical elements; (**c**) EDS line scan along scan path in (**b**); (**d**) HRTEM images; (**e**) Indexed SAED of (**d**).

**Figure 6 materials-15-02214-f006:**
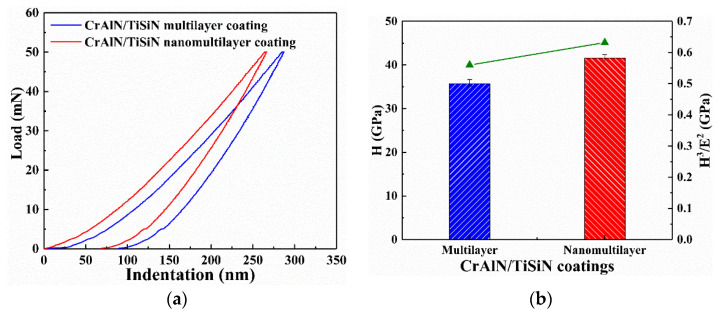
Indentation results of two CrAlN/TiSiN coatings. (**a**) Indentation–load curves; (**b**) Nanohardness and plasticity index.

**Figure 7 materials-15-02214-f007:**
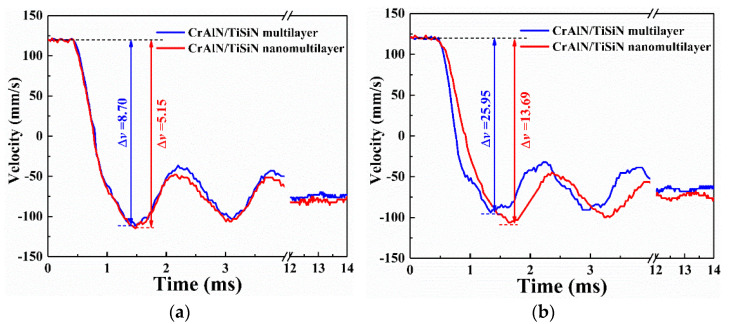
Impact velocity response of two CrAlN/TiSiN coatings at 10^4^ cycles. (**a**) No sand; (**b**) Sand.

**Figure 8 materials-15-02214-f008:**
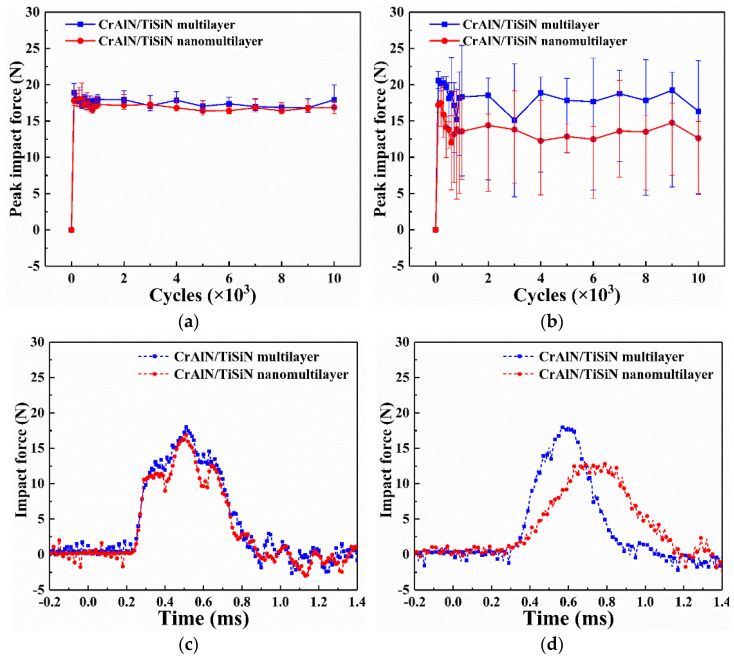
Impact force response of the two CrAlN/TiSiN coatings under various conditions. (**a**) No sand; (**b**) Sand; (**c**) No sand at 10^4^ cycles; (**d**) Sand at 10^4^ cycles.

**Figure 9 materials-15-02214-f009:**
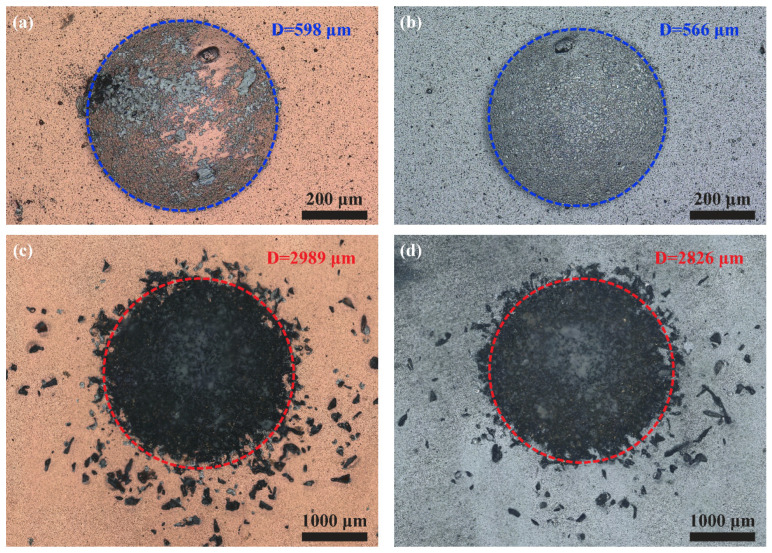
Surface morphology of wear scar: (**a**,**b**) No sand, CrAlN/TiSiN multilayer and nanomultilayer, respectively; (**c**,**d**) Sand, CrAlN/TiSiN multilayer and nanomultilayer, respectively.

**Figure 10 materials-15-02214-f010:**
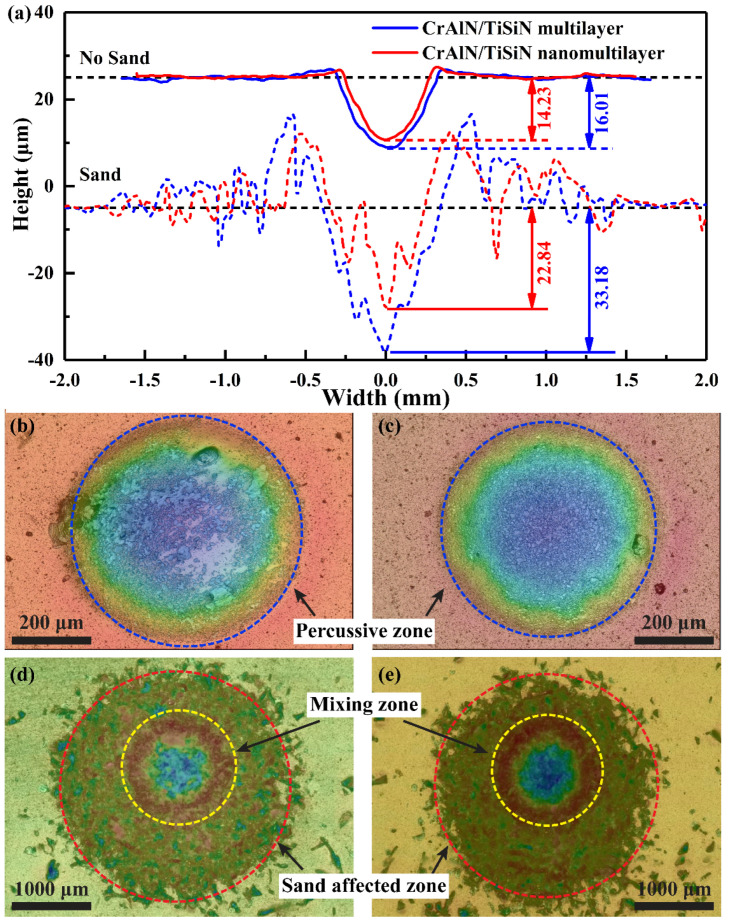
Wear profiles and surface cloud chart of two coatings with and without sand: (**a**) Wear scar profiles; (**b**,**d**) CrAlN/TiSiN multilayer with and without sand; (**c**,**e**) CrAlN/TiSiN nanomultilayer with and without sand.

**Figure 11 materials-15-02214-f011:**
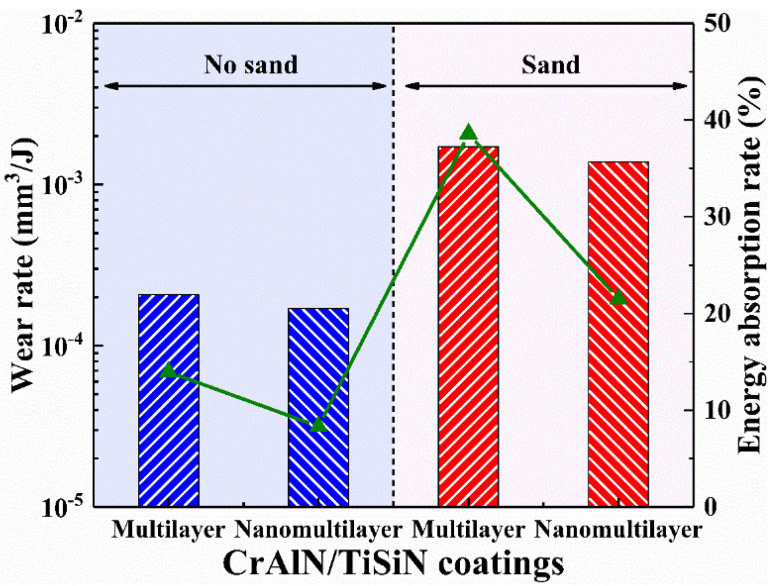
Wear rate and energy absorption rate of two coatings with and without sand.

**Figure 12 materials-15-02214-f012:**
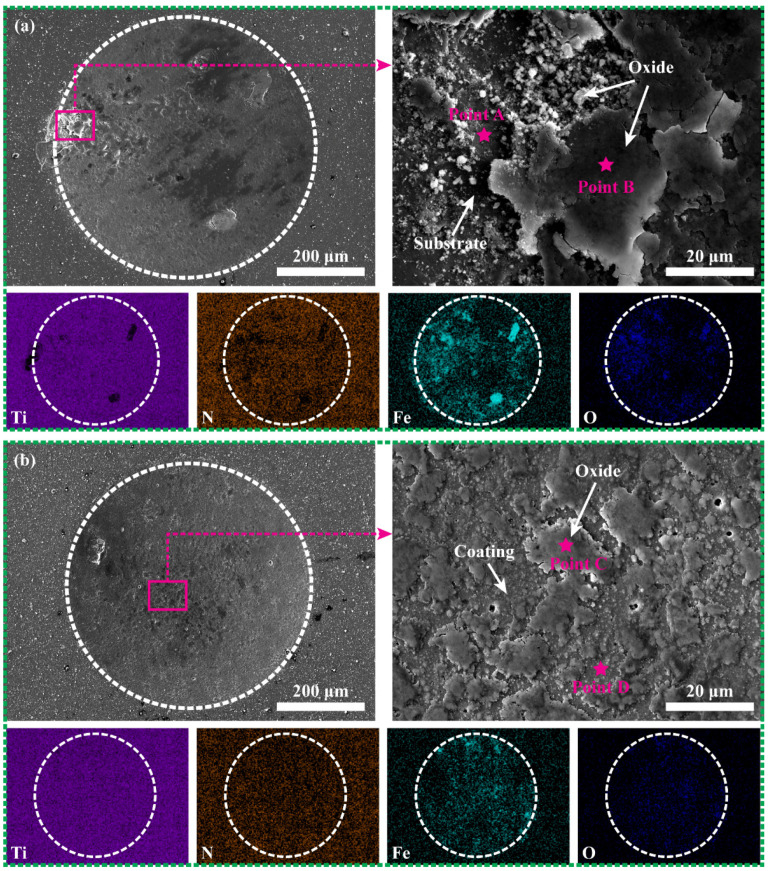
Wear scar morphology and elemental distribution of CrAlN/TiSiN under no−sand conditions: (**a**) Multilayer and (**b**) Nanomultilayer.

**Figure 13 materials-15-02214-f013:**
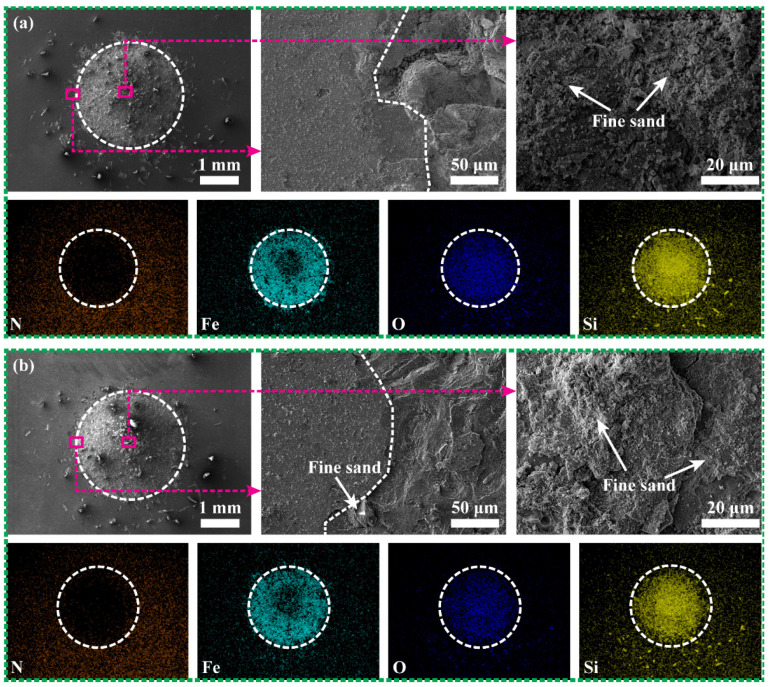
Wear scar morphology and elemental distribution of CrAlN/TiSiN under sand conditions: (**a**) Multilayer and (**b**) Nanomultilayer.

**Figure 14 materials-15-02214-f014:**
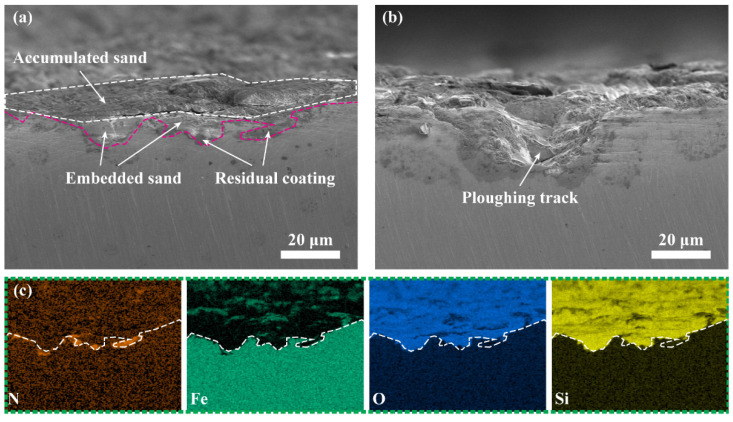
Cross–sectional morphology of nanomultilayer with sand: (**a**,**b**) Typical feature in mixing zone and sand−affected zone, respectively; (**c**) EDS mapping in (**a**).

**Table 1 materials-15-02214-t001:** Detailed parameters of impact wear.

Impact Velocity	Mass Block	Temperature	Cycle	Condition
120 mm/s	200 g	500 °C	10^4^	No sand
				Sand

**Table 2 materials-15-02214-t002:** Chemical composition of the two coatings (wt.%).

Coatings	Ti	Si	Cr	Al	N
CrAlN/TiSiN multilayer	40.68	1.58	37.59	8.13	12.01
CrAlN/TiSiN nanomultilayer	36.01	1.22	41.56	10.58	10.64

**Table 3 materials-15-02214-t003:** Chemical composition at typical position of wear scar (wt.%).

Point	Ti	Si	Cr	Al	N	Fe	O
A	0.4	0.0	46.2	21.9	14.9	14.7	0.0
B	22.8	2.1	22.0	10.8	6.3	21.8	13.5
C	25.3	2.4	27.6	10.1	6.8	10.0	16.7
D	31.2	1.6	36.1	14.6	16.0	0.0	0.0

## Data Availability

The data presented in this study are available on request from the corresponding author.

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
