# Peer review of "Impact Abrasive Wear Property of CrAlN/TiSiN Multilayer Coating at Elevated Temperatures"

_materials, 2022, doi:10.3390/ma15062214_

Round 1

Reviewer 1 Report

See the Enclosure

Reviewer 2 Report

The authors have discussed the abrasive wear properties of CrAlN/TiSiN multilayer and nano-multilayer coatings deposited via multi-arc PVD. The experimentation is nicely designed and the results are adequate to reach to the conclusion. The flow is good and therefore this work is advised for publication. Only a few minor corrections are required:

  1. Include the novelty statement in the introduction section.
  2. Is it a 2 target or multi-target coating system? What's the make?
  3. What were the bias voltages during multilayer and nano-multilayer coating deposition?
  4. Authors mention that the transition layer contains only CrN. According to EDS in fig 4b, the transition layer should contain both CrN and AlN. Can authors double check that and include EDS map of Al in both Fig 4a and 4c?
  5. In description of figs 13 and 14, the right terminology would be 'abrasion', not 'micro-cutting'. Same goes for conclusion section.
  6. line 303: it is not 'attracted', it should be 'attached'.
